# Ancient Mitogenomes Suggest Stable Mitochondrial Clades of the Siberian Roe Deer

**DOI:** 10.3390/genes13010114

**Published:** 2022-01-08

**Authors:** Miao-Xuan Deng, Bo Xiao, Jun-Xia Yuan, Jia-Ming Hu, Kyung Seok Kim, Michael V. Westbury, Xu-Long Lai, Gui-Lian Sheng

**Affiliations:** 1School of Environmental Studies, China University of Geosciences, Wuhan 430078, China; dengmiaoxuan@cug.edu.cn; 2School of Earth Science, China University of Geosciences, Wuhan 430078, China; mr.shaw@outlook.com (B.X.); hujiaming@cug.edu.cn (J.-M.H.); xllai@cug.edu.cn (X.-L.L.); 3State Key Laboratory of Biogeology and Environmental Geology, China University of Geosciences, Wuhan 430078, China; yuanjx@cug.edu.cn; 4Faculty of Materials Science and Chemistry, China University of Geosciences, Wuhan 430078, China; 5Department of Ecology, Evolution, and Organismal Biology, lowa State University, Ames, IA 77575, USA; kkssky@gmail.com; 6GLOBE Institute, Faculty of Health and Medical Sciences, University of Copenhagen, Øster Voldgade 5-7, 1353 Copenhagen, Denmark; m.westbury@sund.ku.dk

**Keywords:** roe deer, ancient DNA, mitochondrial genome, evolutionary history, population dynamics

## Abstract

The roe deer (*Capreolus* spp.) has been present in China since the early Pleistocene. Despite abundant fossils available for detailed morphological analyses, little is known about the phylogenetic relationships of the fossil individuals to contemporary roe deer. We generated near-complete mitochondrial genomes for four roe deer remains from Northeastern China to explore the genetic connection of the ancient roe deer to the extant populations and to investigate the evolutionary history and population dynamics of this species. Phylogenetic analyses indicated the four ancient samples fall into three out of four different haplogroups of the Siberian roe deer. Haplogroup C, distributed throughout Eurasia, have existed in Northeastern China since at least the Late Pleistocene, while haplogroup A and D, found in the east of Lake Baikal, emerged in Northeastern China after the Mid Holocene. The Bayesian estimation suggested that the first split within the Siberian roe deer occurred approximately 0.34 million years ago (Ma). Moreover, Bayesian skyline plot analyses suggested that the Siberian roe deer had a population increase between 325 and 225 thousand years ago (Kya) and suffered a transient decline between 50 and 18 Kya. This study provides novel insights into the evolutionary history and population dynamics of the roe deer.

## 1. Introduction

Roe deer (*Capreolus* spp.) are the only embryonic diapause species in even-toed ungulates. Based on the fossil record, they first appeared about 3.3–2.4 million years ago (Ma) in Udunga, eastern Russia [1], while molecular data suggests an origin around 4–2 Ma [2]. *Capreolus* contains two extant species: the European roe deer (*C. capreolus*) that inhabits most regions of Europe [3], and the Siberian roe deer (*C. pygargus*) that is distributed in the Palaearctic of Asia and parts of Eastern Europe [4].

In a molecular sense, the Siberian roe deer is less studied than the European roe deer and its intraspecific taxonomy is still debated. To date, based on cytogenetic characteristics and morphological parameters, six subspecies of the Siberian roe deer are defined: *C. p. pygargus*, *C. p. tianschanicus*, *C. p. ochracea*, *C. p. bedfordi*, *C. p. mantschuricus*, and *C. p. melanotis* [5,6], the later three of which being somewhat controversial [3,7,8,9,10]. Nevertheless, molecular analyses based on different markers or specimens from different regions suggest various taxonomic statuses of the Siberian roe deer, with at least two or three phylogroups [2,11,12]. Moreover, connections between haplogroups and geographic regions have been investigated in later studies [6,13]. Although there are reports that Siberian roe deer present no apparent phylogeographic structure [14], a recent study by Lee et al. [10] based on partial mitochondrial genes revealed that the geographical distribution of the detected four haplogroups coincide with the previously described ranges of the subspecies: *C. p. pygargus* (to the west of Lake Baikal) and *C. p. tianschanicus* (to the east of Lake Baikal. Additionally, Lee et al. [10] suggested that all haplogroups might have simultaneously originated from an unknown basal haplotype, while further interpretations regarding to the temporal and spatial structures of the Siberian roe deer evolutionary remain speculative.

The roe deer has been present in China since the Early Pleistocene [15,16]. Despite an abundance of Chinese roe deer fossils for detailed morphological analyses, little is known about fossil individuals of this species, such as their taxonomic statuses and relationships with contemporary roe deer. So far, all molecular studies on Chinese roe deer have focused on modern specimens and, to our knowledge, no genetic study using ancient DNA has been performed on Chinese roe deer.

In this study, we generated four ancient mitochondrial genomes from roe deer fossil specimens collected from Northeastern China. We investigated the relationship and divergent events in terms of temporal and geographic view among genetic lineages of roe deer. We also reconstructed the demographic dynamics of this species. The new findings presented in this study provide us further insights into the evolutionary history of the Siberian roe deer.

## 2. Materials and Methods

### 2.1. Samples

Four roe deer sub-fossil specimens (CADG573, CADG580, CADG626, and CADG842) were collected from Qinggang County, Suihua City, Heilongjiang Province, Northeastern China (Figure 1 and Appendix A). All four samples were accelerator mass spectrometry (AMS)-radiocarbon dated at the Beta Analytic Testing Laboratory in the US (Beta-561,491, 561,492, 560,182, and 604,140). The median ages of CADG573, 580, 626, and 842 were 9478, 11,270, 6816, and 9065 calBP (calibrated year before present), respectively (Appendix A).

### 2.2. DNA Extraction

Ancient DNA extractions were performed in a dedicated ancient DNA laboratory at the China University of Geosciences (Wuhan), physically separated from post-PCR facilities. Approximately 200 mg of bone or tooth powder of each sample was digested in extraction buffer which contained 4.5 mL of EDTA (0.5 M, pH = 8) and 0.06 mL of Proteinase K (20 mg/mL), followed by incubation for 16 h in a rotating hybridization oven at 37 °C. After centrifugation at 7000 rpm for 10 min, the supernatant was transferred into an ultrafiltration tube (Millipore, Darmstadt, Germany) and condensed to 100 μL. Finally, DNA was purified and eluted in 80 μL EB buffer using the MinElute PCR Purification Kit (Qiagen, Hilden, Germany) following the manufacturer’s instructions.

### 2.3. Library Construction

Multiple double-stranded Illumina libraries were constructed for each sample using 20 μL DNA extract and the protocol from Meyer and Kircher [23]. We used NEB buffer 2 (New England Biolabs, Ipswich, UK), ATP (New England Biolabs, Ipswich, UK), BSA (New England Biolabs, Ipswich, UK), T4 DNA polymerase (New England Biolabs, Ipswich, UK), T4 polynucleotide kinase (New England Biolabs, Ipswich, UK), and dNTP (Tiangen, Beijing, China) in a blunt-end repair reaction system. In the Adapter Ligation Step, we ligated a 1:20 adapter dilution (Sangon Biotech, Shanghai, China), diluted using Quick Ligase buffer (New England Biolabs, Ipswich, UK) to the ends of DNA fragments. Additionally, we performed an adaptor fill-in reaction using Isothermal buffer (New England Biolabs, Ipswich, UK). Indexing PCR amplifications were prepared with Q5 Hot start High-Fidelity 2× Mater Mix (New England Biolabs, Ipswich, UK) using the following cycling protocol: 98 °C for 30 s and 17 cycles of: 98 °C for 10 s, 60 °C for 75 s, and 60 °C for 6 min. All purification procedures were performed using the MinElute PCR Purification Kit (Qiagen, Hilden, Germany) following the manufacturer’s protocol. Both extraction and library blanks were performed to monitor for potential contamination. Library quality and concentration were measured using Qubit 4.0 (Invitrogen, Carlsbad, CA, USA) and TapeStation 4150 (Agilent, Santa Clara, CA, USA). Finally, indexed libraries were pooled into a single pool in equimolar ratios and sequenced on an Illumina HiSeq×10 platform at Personalbio Inc, Shanghai, China.

### 2.4. Data Processing and Data Sets

Raw reads were processed to trim Illumina adapter sequences using Cutadapt v1.4.2 [24], discarding reads shorter than 30 bp. Flash v. 1.2.11 [25] was used to merge overlapping read pairs. To avoid loss of reads near the break point, the trimmed reads were then mapped against the complete mitochondrial genome of a modern Siberian roe deer (GenBank accession number: NC_025271) twice with different break points using the “aln” and “samse” algorithms in Burrows-Wheeler Aligner v. 0.6.2 [26] with default parameters. Next, sequences below a mapping quality of 30 were filtered out using “view” and the alignment was sorted on the reference genome by 5′ mapping position using “sort” in SAMtools v. 0.1.19 [27]. In addition, potential PCR duplicates were removed using “rmdup”, and a single bam format file of each library was obtained using “merge” in SAMtools. Finally, mitochondrial consensus sequences were constructed using GENEIOUS v.10 (http://www.geneious.com/, accessed on 25 December 2021) based on the final bam files and considering only bases with a sequencing depth of ≥2×. At every position, a nucleotide was called only if it was observed in ≥ 75% of the reads covering that site [28,29]. Read coverages across the reference were calculated using Qualimap v. 2.2.1 [30] and the evaluation of base damage for ancient DNA was performed using mapDamage2 with default parameters [31].

The four newly retrieved mitochondrial genomes were deposited in GenBank (Accession numbers: OK041025-27 and OK323229). Here, four data sets were defined to initiate different analyses.

Data set 1 comprised 22 complete mitochondrial genomes (Appendix A): 11 Siberian roe deer, nine European roe deer, one Chinese water deer (*Hydropotes inermis*), and one moose (*Alces alces cameloides*) [32,33,34,35,36,37] plus the four newly obtained ancient mitochondrial genomes of our samples. To improve the accuracy of the divergence time estimation, Data set 1 retained the most complete sequences. This data set was used for phylogenetic analyses and molecular dating among roe deer populations.

Data set 2 contained 146 combined sequences of *cyt b* and the D-loop (Appendix A) from samples with detailed geographic information: 141 modern Siberian roe deer (covering Korea, Mongolia, Kazakhstan, Siberia, and Poland) [10,13,33,35,36,37], the four newly obtained fossil Chinese roe deer from this study, and one European roe deer which served as an outgroup. This data set, which had a high number of individuals sampled across most of the geographical range of Siberian roe deer, was used to investigate the phylogeographical structure of Siberian roe deer.

Data set 3 comprised 11 modern complete mitochondrial genomes and our four ancient near-complete mitochondrial genomes of the Siberian roe deer. Data set 4 comprised nine complete mitochondrial genomes of modern European roe deer. Data sets 3 and 4 (Appendix A) were used to reconstruct the maternal demographic histories for the Siberian roe deer and the European roe deer, respectively.

### 2.5. Bioinformatics Analyses

Sequence alignments were carried out using MAFFT [38] on the CIPRES portal [39], and the alignments were checked manually.

We first performed maximum-likelihood (ML) phylogenetic analysis on Data set 1 with RAxML-HPC v. 8 [40] on the CIPRES portal to address the phylogenetic relationships among roe deer. As the D-loop was characterized by high interspecific variation and poorly aligned regions, it was manually eliminated before subsequent analyses. *H. inermis* and *A. alces* served as outgroups to root the tree. Considering the heterogeneity of evolutionary rate among partitions, the best fitting partitioning scheme and substitution models were determined by PartitionFinder2 [41] with the greedy search algorithm and linked branch lengths along with Bayesian Information Criterion, resulting in five partitions with the GTR+I nucleotide substitution model for most of the partitions (Appendix A). For our ML analysis, node support was assessed using 500 bootstrap replicates.

We then performed Bayesian analysis on Data set 1 using BEAST 1.8.4 [42] to investigate the most recent common ancestor (TMRCA) of each lineage of roe deer. The substitution model GTR + I + G was selected through comparison of Bayesian Information Criterion scores in jModelTest v2.1 [43]. In order to calibrate our BEAST analysis, we constrained the age of two nodes based on a previous molecular dating study [44]: the TMRCA of *Capreolus* and *A. alces* with a mean of 13.9 Ma (Normal distribution, 95% interval of 15.89–11.99 Ma) and the divergence between *Capreolus* and *H. inermis* with a mean of 9.48 Ma (Normal distribution, 95% interval of 11.31–7.76 Ma). Meanwhile, we specified a strict molecular clock, and a constant population size in parameter settings. Markov Chain Monte Carlo (MCMC) sampling was carried out using 70 million iterations with sampling every 1000 steps. Runs convergence and the effective sample size (ESS > 200) were checked using Tracer v1.7 [45], discarding the first 10% as burn-in. The maximum clade credibility (MCC) tree was generated using TreeAnnotator v1.8.4 [46] and visualized in FigTree v1.4.3 (http://tree.bio.ed.ac.uk/software/figtree, accessed on 25 December 2021).

Given the limited number of complete mitochondrial genomes, we used Data set 2 (146 combined sequences of *cyt b* and the D-loop with detailed geographic information) to investigate the phylogeographical structure of Siberian roe deer. We grouped the locations of Data set 2 into eight regions (Appendix A) according to geographical proximity as described in Lee et al. [10]: Jeju Island in South Korea (SKJ); South Korea mainland (SKM); Primorsky Krai and the Amur region of Russia and Northeastern China (RPRA); Yakutia in Russia (RYA); the Sokhondinsky Nature Reserve in the Trans-Baikal region of Russia, Buryatia in Russia, and Northern Mongolia (RSMG); Altay, Novosibirsk, Krasnoyarsk, Khakassia, and Irkutsk in Russia and Kazakhstan (RARN); the Urals, Kurgan, Orenburg, and Sverdlovsk in Russia (RUKO); and Poland (PLD). We constructed another phylogenetic tree by Bayesian inference in MrBayes v 3.2.7 [47] on the CIPRES portal to define haplogroups of different geographic populations. The parameters were the same as previous analysis, except for the substitution model, for which we used HKY+I+G as indicated by jModelTest v2.1 [43].

To construct the maternal demographic histories of the Siberian roe deer and the European roe deer, we performed Bayesian skyline plots (BSPs) on Data sets 3 and 4 using BEAST v. 1.8.4 [42]. The skyline plot was time-calibrated using a substitution clock rate with a mean of 8.91 × 10^−9^ substitutions per site per year (normal distribution, 95% HPD: 10.01 × 10^−9^–7.71 × 10^−9^) suggested by our previous Bayesian analysis on Data set 1. Optimal substitution models for two data sets were assigned by jModelTest v2.1 [43], resulting in HKY + I for Data set 3, and HKY + G for Data set 4. Additionally, the age of our ancient samples was set as the mean of the calBP in tip dates. Other parameters were set the same as previously.

## 3. Results

We successfully retrieved four near-complete mitochondrial genomes of the roe deer with mean depths of 12.7, 9.0, 14.5, and 5.9 folds, respectively (Appendix A). The results of mapDamage2 indicated advanced DNA fragmentation and high levels of cytosine deamination at the terminal DNA fragment ends (Appendix A), authenticating that our sequences were not derived from modern contamination.

The phylogenetic trees computed using either RAxML-HPC or BEAST v. 1.8.4 yielded a consistent topology (Figure 2 and Appendix A), which indicated two main clades of roe deer (the European roe deer vs. the Siberian roe deer) as previously suggested [2,33,48,49]. The Siberian roe deer clade is further divided into three distinct subclades, and subclade P3 occupies a slightly earlier diverging position, while subclades P1 and P2 are sister to each other. Interestingly, our four ancient samples fell into two different subclades of the Siberian roe deer, with CADG573 and CADG626 in subclade P2, while CADG580 and CADG842 fell within subclade P3.

Assuming a 13.9 Ma calibration point for *Capreolus* and *A. alces* divergence, and a 9.48 Ma calibration point for *Capreolus* and *H. inermis* divergence, our dated Bayesian phylogenetic analysis based on near-complete modern and ancient mitogenomes (Figure 2 and Appendix A) revealed that the Siberian roe deer and the European roe deer coalesce at 2.25 Ma (95% HPD: 2.63–1.87 Ma), which is consistent with previous molecular results (4–2 Ma) [2,48]. The TMRCA for the European roe deer clades was estimated at 0.68 Ma (95% HPD: 0.81–0.55 Ma), while TMRCA for the Siberian roe deer clades was calculated at 0.34 Ma (95% HPD: 0.41–0.27 Ma). The subclades P1, P2 and P3 diverged at 0.10 Ma (95% HPD: 0.14–0.07 Ma), 0.17 Ma (95% HPD: 0.23–0.14 Ma), and 0.30 Ma (95% HPD: 0.37–0.24 Ma), respectively.

The phylogenetic analysis of Data set 2 (combined *cyt b* and D-loop) divided the Siberian roe deer into four haplogroups (A, B, C, and D) (Figure 3a). The number of haplogroups is different from previous analysis using complete mitochondrial genomes, which indicates only three subclades (Figure 2 and Appendix A). When comparing phylogenetic trees among analyses of Data sets 1 and 2 (Figure 2 and Figure 3a), we observed that the phylogenetic positions of overlapping samples were consistent. We saw that haplogroup B corresponded to subclade P1, haplogroups A and D corresponded to subclade P2, and haplogroup C corresponded to subclade P3. While considering that too few complete mitochondrial genomes may lead to incomplete details of branches, and that the rapid evolutionary rate of D-loop makes it more effective to distinguish individuals with close genetic relationships, the former two results of Data sets 1 and 2 are not contradictory.

To better visualize the phylogeographical structure of the Siberian roe deer, we mapped the geographical distribution of haplogroups (Figure 3b). Haplogroups B and C are geographically extensive, ranging from Central Europe to East Asia (across both sides of Lake Baikal), while haplogroups A and D were only found in the east of Lake Baikal. Populations on both sides of Lake Baikal have different haplogroup compositions. Specifically, except for the isolated population SKJ, most of the individuals in the western population belong to haplogroup B, while all haplogroups exist in the eastern population. Only one haplogroup (haplogroup B) was detected in Jeju Island, and this may be due to geographical isolation resulting in less gene flow between populations in a short period of time or processes such as founder effect and genetic drift as evidenced by reduced genetic diversity of the animal group on the island [50]. Our ancient samples CADG573, CADG580, CADG626, and CADG842 fell into haplogroup D, haplogroup C, haplogroup A, and haplogroup C, respectively (Figure 3a).

The BSPs (Figure 4) revealed changes in maternal effective population size of Siberian and European roe deer over time. The maternal effective population size of European roe deer had a major expansion during ~100 to ~50 thousand years ago (Kya), followed by a relatively stable population size to the present. In contrast, the Siberian roe deer experienced a population increase during ~325 to ~225 Kya and had a transient decline during ~50 to ~18 Kya.

## 4. Discussion

### 4.1. Phylogeny and Phylogeography of the Roe Deer in Northeastern China

Previous molecular identifications have suggested extant Northeastern China roe deer as Siberian roe deer [48,51,52]. However, there is also debate as to whether it is a subspecies or not. Morphologically, it has been assigned to different subspecies of Siberian roe deer, i.e., *C. p. bedfordi* [53], *C. p. tianschanicus* [4], or *C. p. manchuricus* [48]. Further morphological evidence suggested that the subspecies *C. p. bedfordi* and *C. p. manchuricus* are synonyms of *C. p. tianschanicus*, which is distributed in Mongolia, Trans-Baikalia, the Russian Far East, and China [54,55]. Our maximum-likelihood and Bayesian results (Figure 2 and Appendix A) strongly support that the sub-fossil roe deer in Northeastern China belong to the Siberian roe deer, which is in line with previous studies based on D-loop [48,51,52]. Moreover, we observed similar genetic composition in all populations to the east of Lake Baikal (Figure 3b), including Northeastern China. Therefore, it seems reasonable to define Northeast Chinese roe deer as *C. p. tianschanicus*.

In the phylogenetic trees inferred from Data set 1 (Figure 2 and Appendix A) our four ancient samples, dated to between 6816–11,270 calBP, fell into two different subclades of modern Siberia roe deer, i.e., subclades P2 and P3, indicating genetic continuity for this species in Northeastern China, at least since the end of the Late Pleistocene to the present. Furthermore, we observed that subclade P2 contained both ancient (CADG573: 6816 calBP and CADG626: 9478 calBP) and modern Northeastern China roe deer samples (MN485773), and we thus speculate that subclade P2 has existed in Northeastern China throughout the early Holocene into the present. Our phylogenetic analyses using Data set 2 (Figure 3) revealed that our four ancient samples, grouped with haplotypes from RPRA, distribute in three different haplogroups, i.e., haplogroup A, haplogroup C, and haplogroup D. These results demonstrate that haplogroup C, distributed throughout Eurasia today, was in Northeastern China since at least the Late Pleistocene, while haplogroup A and haplogroup D, only distributed in the east of Lake Baikal today, were distributed in Northeastern China since the Middle Holocene. Moreover, both phylogenetic analyses (Figure 2, Figure 3a and Appendix A) support an initial divergence of haplogroup D/subclade P3, which indicates that haplogroup D/subclade P3 may represent the oldest genetic lineage.

Similar to previous studies [10,12,13,14], our geographical distribution of haplogroups (Figure 3b) showed that Siberian roe deer from Central Europe to East Asia are highly heterogeneous without clear geographical separation of maternal lineages. This feature might be derived from substantial gene flow due to the high mobility of this species [9,56], as well as high levels of connectivity among populations during the Ice Age [33]. Furthermore, we found that taking Lake Baikal as the boundary, except the isolated population SKJ, the western populations mainly belong to haplogroups B and C, while all haplogroups are found in the eastern populations, suggesting a different haplogroup composition of populations on both sides of Lake Baikal. Consistent with previous studies [10,50,57], this result supports the existence of the subspecies *C. p. pygargus* (to the west of Lake Baikal) and *C. p. tianschanicus* (to the east of Lake Baikal) and emphasizes the role of Lake Baikal as a geographic barrier to gene flow. In contrast to Lee et al. [10], we detected haplotypes of haplogroup B in SKM, probably because our data set contains more samples that make new haplogroup discovery possible. Moreover, we observed that the SKM, RSMG, and RPRA populations were distributed in all four haplogroups, suggesting high mitochondrial genetic diversity in these regions. It may indicate either a long term large effective population size or East Asia is the center of Siberian roe deer diversification. In East Asia, several vertebrate species have also been reported to possess high levels of mitochondrial DNA variations compared with individuals from surrounding areas [58]. The co-existence of cold open steppes and forests in East Asia could provide favorable conditions for a large population and thus the diversification of genetic lineages [50,59].

### 4.2. Divergence among Capreolus

Currently, the oldest record of *Capreolus* is from the Late Pliocene (MN16, 3.3–2.4 Ma), discovered in Udunga, eastern Russia [1]. Previous molecular studies based on partial mitochondrial genes have estimated the separation time of the European roe deer and the Siberian roe deer at around 4–2 Ma [2,48]. A recent study using the nuclear genomes of modern individuals of Siberian roe deer in Northeastern China and European roe deer in Germany estimated that these two species diverged 1.35–0.9 Ma [60]. Our estimation (2.25 Ma; 95% HPD: 2.63–1.87 Ma) is compatible with previous studies based on partial mitochondrial genes but is, however, earlier than the estimation using nuclear genes. We proposethat the discordance may be caused by nuclear gene flow. Moreover, according to the fossil record [61], the European roe deer was already present in Europe at least 0.6 Ma. Our estimated coalescence of the European roe deer (0.68 Ma; 95% HPD: 0.81–0.55 Ma) conforms to the fossil record. In addition, using D-loop data, Randi et al. [2] showed that the divergence among haplotypes of the Siberian roe deer might have arisen 0.37–0.19 Ma and we obtained similar coalesce at 0.34 Ma (95% HPD: 0.41–0.27 Ma). Among the Siberian roe deer, we found the coalescence events of subclades P1 and P2 occurred close in time (~0.10 Ma vs. ~0.18 Ma). This comparatively rapid divergence might be explained by glacial cycles, which would result in range shifts and geographical isolation in refugia of the Siberian roe deer and further induce inter-specific diversification.

### 4.3. Maternal Demographic History of the Roe Deer

Using nuclear genes, de Jong et al. [60], suggested that the Siberian roe deer experienced two increases in effective population size from ~2000 to ~200 Kya and ~140 to ~50 Kya, and suffered a sharp decrease from ~30 to ~18 Kya. In our BSPs (Figure 4), the maternal population size of the Siberian roe deer increased gradually from ~325 Kya during the Middle Pleistocene and peaked at around 225 Kya, during the Marine Isotope Stage (MIS) 7 interglaciation (243–191 Kya), followed by a stabilization period from ~225 Kya to ~50 Kya. Subsequently, the maternal population size decreased by half from ~50 Kya to ~18 Kya, and then remained constant until present. This different pattern of expansions could be caused by episodes of gene flow in the absence of changes in population size.

With regard to the European roe deer, Randi et al. [62] identified two waves of population expansion from ~244 to ~122 Kya and ~156 to ~78 Kya, respectively, using the mismatch distributions of the D-loop. In contrast, our BSP analysis (Figure 4) based on a near-complete mitochondrial genome indicated one apparent population expansion from ~100 to ~50 Kya. This inconsistency may have resulted from both different datasets with different evolution rates, and different sampling schemes in which we excluded samples from locations where near complete mitochondrial sequences are not available.

For the increase in the maternal population size, both a demographic expansion and a population subdivision have been proposed as possible reasons [63]. The increase in the maternal population size of the Siberian roe deer occurred around 0.325 Ma, which is consistent with the divergence of subclade P3 (0.30 Ma; 95% HPD: 0.37–0.24 Ma) (Figure 2 and Appendix A); the increase in the maternal population size of the European roe deer appeared around 0.10 Ma, which also corresponds to the emergence of subclade C3 of the European roe deer (0.13 Ma; 95% HPD: 0.18–0.08 Ma). Therefore, we hypothesise that the increase in the maternal effective population size of both roe deer species may be the consequences of population subdivisions rather than population expansions. Moreover, global climate change and human activities likely played a key role in the observed population size shrink of the Siberian roe deer. The identified period of bottleneck (from ~50 Kya to ~18 Kya) coincides with the transition from the climatically unstable MIS 3 to the more stable cold period of MIS 2 around 29 Kya. Similar to other species from temperate climates, the Siberian roe deer expanded during interglacials and contracted into cryptic and isolated refugia during glacials [62,64]. Numerous archaeology discoveries indicated that this period contained frequent human activities in Eurasia [65]; thus, human activities may have had a negative impact on the population size of the Siberian roe deer.

## 5. Conclusions

In summary, our study provides real-time molecular records for tracing the divergence and demographic history within roe deer populations. We confirmed that the ancient roe deer shares a close genetic background with modern roe deer, which implies that this species maintains stable mitochondrial clades since the Late Pleistocene even though it has experienced multiple dynamics in its population size. In addition, our results also revealed the connection between climate change and the evolutionary events of roe deer. The identified period of bottleneck coincides with the transition of unstable climatic stage to more stable cold period, which suggests that the cooling periods might have resulted in extensive decrease of population size of roe deer.

## Figures and Tables

**Figure 1 genes-13-00114-f001:**
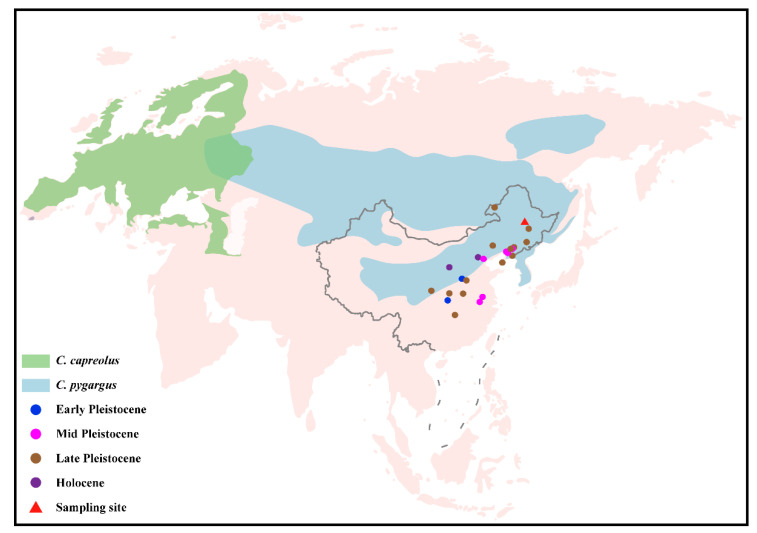
Distribution ranges of two roe deer species (*C. capreolus* and *C. pygargus*) [5]. The sampling site in this study is indicated by a red triangle. Other roe deer fossils are represented by dots in four different colors [17,18,19,20,21,22]. The base map of global world (chart No. GS (2016)1566) and China (chart No. GS (2019)1686) were downloaded from the National Administration of Surveying, Mapping and Geoinformation of China (http://bzdt.ch.mnr.gov.cn, accessed on 19 May 2021).

**Figure 2 genes-13-00114-f002:**
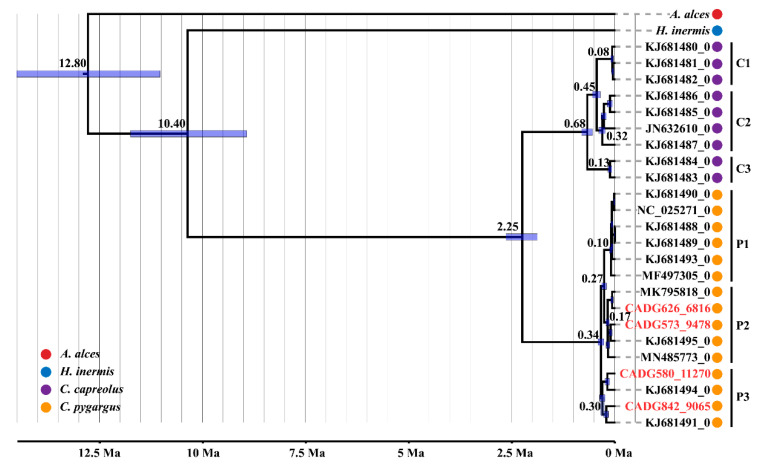
Maximum clade credibility tree of *Capreolus* computed in BEAST based on complete mitogenomes. Node heights are centered on the median posterior age estimates (*X*-axis; Ma) and blue node bars show 95% credibility intervals of the divergence times. Tip dates of specimens are indicated following the accession/sample number. The four ancient samples from this study are labelled in red letters.

**Figure 3 genes-13-00114-f003:**
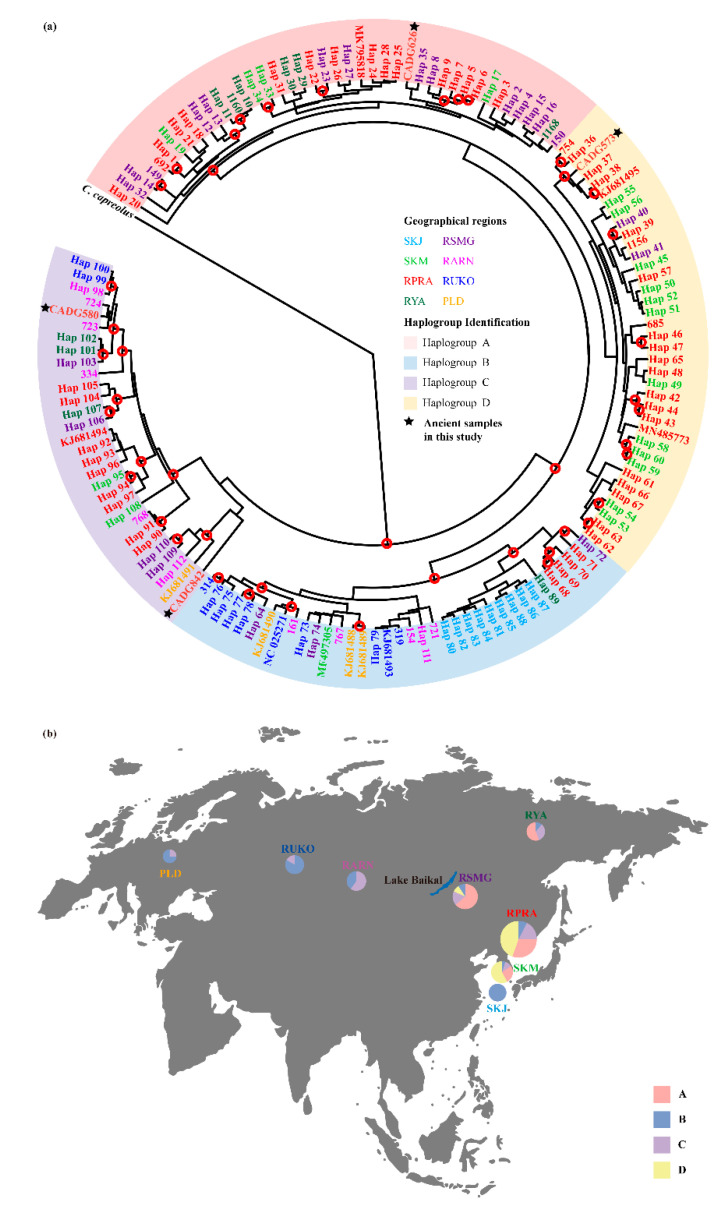
Phylogeographic distribution and Bayesian phylogenetic tree of the Siberian roe deer specimens with precise geographic localities (Data set 2, Appendix A). (**a**) Bayesian phylogenetic tree of Siberian roe deer based on combined sequences of *cyt b* and the D-loop. Red hollow circles denote nodes supported by posterior probabilities over 95%. Tip labels are colored based on haplogroup. Colored letters correspond to different geographic regions that are markedin the same color. The four ancient samples from this study are shown by black stars. (**b**) Geographical distribution of the haplogroups revealed by Bayesian analysis. Circles are sized relative to frequency of the respective phylogenetic haplogroups (A, B, C, D). Lake Baikal is shown in blue in the bottom middle of the map. Regional abbreviations: Jeju Island in South Korea (SKJ); South Korea mainland (SKM); Primorsky Krai and Amur region of Russia and Northeastern China (RPRA); Yakutia in Russia (RYA); the Sokhondinsky Nature Reserve in the Trans-Baikal region of Russia, Buryatia in Russia, and Northern Mongolia (RSMG); Altay, Novosibirsk, Krasnoyarsk, Khakassia, and Irkutsk in Russia and Kazakhstan (RARN); the Urals, Kurgan, Orenburg, and Sverdlovsk in Russia (RUKO); Poland (PLD). The base map of global world (chart No. GS (2016)1566) was downloaded from the National Administration of Surveying, Mapping and Geoinformation of China (http://bzdt.ch.mnr.gov.cn, accessed on 19 May 2021).

**Figure 4 genes-13-00114-f004:**
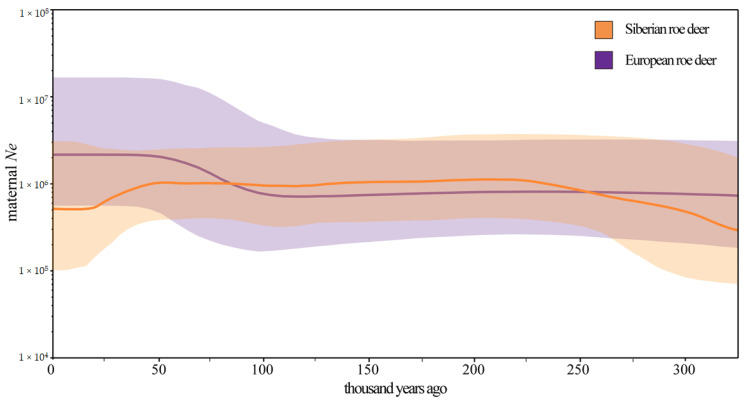
Maternal demographic histories of Siberian and European roe deer reconstructed from mitochondrial genomes. *Y*-axis represents the female effective population size (*Ne*) while *X*-axis stands for time. Lines in different colors indicate median female *Ne* and the corresponding shades show the 95% credibility interval.

## Data Availability

Four ancient mitochondrial genomes have been submitted to the GenBank database under accession number OK041025 (CADG573), OK041026 (CADG580) OK041027 (CADG626) and OK323229 (CADG842). The data will be released on 8 September 2022. The address is as follows: GenBank www.ncbi.nlm.nih.gov/genbank, accessed on 3 January 2022.

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
