# Peer review of "Ancient Mitogenomes Suggest Stable Mitochondrial Clades of the Siberian Roe Deer"

_genes, 2022, doi:10.3390/genes13010114_

Round 1

Reviewer 1 Report

The authors have presented a well designed, carried out, and described phylogenetic and population genomic study on the Roe Deer. This is the first aDNA study of Chinese Roe Deer to observe their relationship to modern Roe Deer and placement within the Capreolus phylogenetic tree. They generate four ancient mitochondrial genomes from Roe Deer bones collected in China. Background is well laid out with references to previous literature and lays the basis for the merit of this paper. Methods section is well described with full details of DNA processing. There are some basic details missing regarding the amount of data generated and aligned, as well as basic statistics on endogenous content and fragment length. Overall, the most attention is needed to English language. This study is well designed and presented.

English grammar needs attention throughout entirety of text. Some examples of corrections from the first couple of paragraphs are below. This is not a comprehensive list.  

-Line 17: The roe deer (sp) *has been* present in China since the Early Pleistocene

-Line 18: Despite abundant fossils are available for detailed morphological analyses

-Line 22: Phylogenetic analyses indicate *the*four ancient

-Line 36: “Based on the fossil record, the roe deer” first appeared …

-Line 40:  that *is* distributed* across the Asian and Eastern European Palaearctic

Comments on Figures

-figure 1a pink and yellow are hard to differentiate. Change colors.

-figure 1b “Sampling sites” in legend should just be “sampling site” as there is only one on the map.

Methods:

-section 2.3- What type of adapter? Nextera? Truseq?

-line 113. How many reads per sample were targeted in sequencing?

Results:

-how many sequences were generated per sample?

-how many sequences aligned per sample?

-What were the endogenous contents and average fragment lengths?

Reviewer 2 Report

I reviewed the article titled: “Ancient mitogenomes suggest stable mitochondrial clades of the  Siberian roe deer” and I found it very interesting and well-prepared. The authors generated four mitogenomes obtained from bones of roe deer and carried out phylogenetic analyses in order to determine the evolutionary history of this species. In my opinion the article written by Miao-Xuan Deng et al. might be published in Genes after minor revision. Please refer to the following points:

Abstract:

[27] The abbreviations Ma and Kya should be explained at this point as the abstract is a separate part of the article.

Introduction:

[41] The authors indicate that Siberian roe deer is less molecularly studied - could you explain it in order to enrich the introduction section?

[60] The roe deer has been already present in China since the Early Pleistocene

[64] are focused on

Materials and Methods:

[74] The authors analysed four fossils. Is it possible to determine what bones were?  

[89] Did the bones or teeth were mechanically grind prior to DNA isolation?

[95] Did the authors verify the quantity and/or quality after isolation?

[115] Did the authors take into consideration a possible NUMT coamplification?

[125] What was the minimal and maximal read coverage?

[130] There is no information in GenBank for these accession numbers - the authors should state when the data is going to be released
